# Molecular biomarkers in Batagay megaslump permafrost deposits reveal clear differences in organic matter preservation between glacial and interglacial periods

Loeka L. Jongejans[1,2], Kai Mangelsdorf[3], Cornelia Karger[3], Thomas Opel[4], Sebastian Wetterich[1,5], Jérémy Courtin[2,4], Hanno Meyer[4], Alexander I. Kizyakov[6], Guido Grosse[1,2], Andrei G. Shepelev[7], Igor I. Syromyatnikov[8], Alexander N. Fedorov[7] and Jens Strauss[1]

[1]Alfred Wegener Institute Helmholtz Centre for Polar and Marine Research, Permafrost Research Section, 14473 Potsdam, Germany

[2]Institute of Geosciences, University of Potsdam, 14476 Potsdam, Germany

[3]GFZ German Research Centre for Geosciences, Section Organic Geochemistry, 14473 Potsdam, Germany

[4]Alfred Wegener Institute Helmholtz Centre for Polar and Marine Research, Polar Terrestrial Environmental Systems Section, 14473 Potsdam, Germany

[5]Current address: Technische Universität Dresden, Institute of Geography, 01069 Dresden, Germany

[6]Cryolithology and Glaciology Department, Faculty of Geography, Lomonosov Moscow State University, 119991 Moscow;

[7]Laboratory of Permafrost Landscapes, Melnikov Permafrost Institute, Siberian Branch of the Russian Academy of Science, 677010 Yakutsk, Russia

[8]Laboratory of General Geocryology, Melnikov Permafrost Institute, Siberian Branch of the Russian Academy of Science, 677010 Yakutsk

*Correspondence to*: Loeka L. Jongejans (loeka.jongejans@awi.de), second contact Jens Strauss (jens.strauss@awi.de)

**Abstract.** The Batagay megaslump, a permafrost thaw feature in north-eastern Siberia, provides access to ancient permafrost up to ~650 ka old. We aimed to assess the permafrost-locked organic matter (OM) quality and to deduce palaeo-environmental information on glacial-interglacial timescales. We sampled five stratigraphic units exposed on the 55 m high slump headwall and analysed lipid biomarkers (alkanes, fatty acids and alcohols). Our findings revealed similar biogeochemical signatures for the glacial periods: the Lower Ice Complex (Marine Isotope Stage (MIS) 16 or earlier), the Lower Sand Unit (sometime between MIS 16-6) and the Upper Ice Complex (MIS 4-2). The OM in these units has a terrestrial character, and microbial activity was likely limited. Contrarily, the *n*-alkane and fatty acid distributions differed for the units from interglacial periods: the Woody Layer (MIS 5), separating the Lower Sand and the Upper Ice Complex, and the Holocene Cover (MIS 1), on top of the Upper Ice Complex. The Woody Layer, marking a permafrost degradation disconformity, contained markers of terrestrial origin (sterols) and high microbial decomposition (*iso-* and *anteiso-*fatty acids). In the Holocene Cover, biomarkers pointed to wet depositional conditions and we identified branched and cyclic alkanes, which are likely of microbial origin. Higher OM decomposition characterised the interglacial periods. As climate warming will continue permafrost degradation in the Batagay megaslump and in other areas, large amounts of deeply buried, ancient OM with a variable composition and degradability are mobilised, likely significantly enhancing greenhouse gas emissions from permafrost regions.

## 1 Introduction

Rapid warming of the terrestrial Arctic leads to widespread permafrost thaw. This can mobilise organic matter (OM) and results in greenhouse gas (GHG) release, which contributes to the permafrost-carbon climate feedback (Schuur et al., 2015). The global permafrost region contains roughly half of the world's soil carbon (3350 Gt), and in addition, a large deep permafrost carbon pool (>3 m) which is often not accounted for and its amount is uncertain (~500 Gt) (Strauss et al., 2021). While it was estimated that gradual permafrost thaw might contribute up to 208 Gt carbon into the atmosphere until 2300

(McGuire et al., 2018), abrupt permafrost thaw processes, such as the formation of retrogressive thaw slumps and thermokarst development, could contribute an additional $80 \pm 19$ Gt of carbon to be released into the atmosphere (Turetsky et al., 2020). Abrupt thaw processes occur on local to regional scales and are difficult to capture, which is why they are not yet implemented in climate models.

Retrogressive thaw slumps are a result of slope failure following the thaw of ice-rich permafrost. They develop rapidly and

can displace large amounts of ice/water, sediments and OM (Lewkowicz, 1987; Lantuit and Pollard, 2005; Tanski et al., 2017). Thaw slumps typically consist of a nearly vertical headwall, a slump floor and a lobe, and are often situated along rivers or coasts. Triggers for the formation include lateral or thermal erosion by water (Kokelj et al., 2013), active layer detachment following heavy rainfall (Lacelle et al., 2010) and human activity such as road construction, mining or deforestation. Once initiated, thaw slumps can develop very rapidly due to the constant removal of thawed material by meltwater streams, changes

in the vegetation and snow cover, and the albedo leading to further intense permafrost degradation.

The Batagay megaslump in East Siberia is the largest known retrogressive thaw slump on Earth (roughly 1.8 km long and 0.9 km wide in 2019) that developed over the last ~5 decades (Kunitsky et al., 2013). The megaslump provides access to ancient permafrost deposits, with stratigraphical discordances, including the second oldest directly dated permafrost in the Northern Hemisphere (Murton et al., 2022). This makes the large slump headwall an ideal target for palaeo-environmental studies,

including cryostratigraphy, sedimentology and chronology (Ashastina et al., 2017; Murton et al., 2017, 2022), ground ice stable isotopes (Opel et al., 2019; Vasil'chuk et al., 2020), pollen and plant macro remains (Ashastina et al., 2018) and ancient DNA (Courtin et al., accepted).

The study of lipid biomarkers has been proven useful in previous work to characterise permafrost OM and carbon cycling as well as tracing permafrost thaw (Zech et al., 2009; Strauss et al., 2015; Elvert et al., 2016; Stapel et al., 2016; Jongejans et al.,

2018, 2020; Martens et al., 2020; Bröder et al., 2021; Yao et al., 2021). With the present study we aim (1) to explore the source and preservation of biomarkers in permafrost on geologic time scales during several glacial and interglacial periods, and (2) to deduce the past floral and microbial sources of the still preserved OM in order to characterise palaeo-environments of OM deposition. To our knowledge, we present the first OM signatures, i.e. biomarkers preserved in ancient permafrost since about 650 ka.

## 2 Study site

The Batagay megaslump (67.58°N, 134.77°E) close to the village of Batagay is located in the Yana Uplands, part of Yana-Oymyakon mountain region (interior Yakutia; Figure 1a). This region is characterised by the most continental climatic conditions of the Northern Hemisphere, manifesting in an extreme climate with a mean winter (December to February) temperature of -40.0°C, a mean summer (July to August) temperature of 13.7°C and a mean annual temperature of -12.4°C (period 1988-2017) (Murton et al., accepted). For the same time period, mean annual precipitation was 203 mm, with mean summer precipitation of 106 mm. Since the mid-20th century, both temperature and precipitation have significantly increased. The permafrost in this region is continuous and ~200 to 500 m thick with mean annual ground temperatures of -8.0 to -5.5°C (Murton et al., accepted). The seasonally thawed uppermost (active) layer is between 0.2 and 1.2 m thick, depending on vegetation type (Murton et al., accepted). The modern vegetation is dominated by open larch forest (*Larix gmelinii*), and Siberian dwarf pines (*Pinus pumila*) and birch trees (*Betula exilis*, *B. divaricata* and sparse *B. pendula*) are common. The ground is covered by a thick layer of lichens and mosses, and almost no grasses and herbs are present (Ashastina et al., 2018; Murton et al., 2022).

The Batagay megaslump is located on an east-facing hillslope and has developed after anthropogenic disturbance of the protective vegetation cover in the middle of the 20th century (Kunitsky et al., 2013; Murton et al., accepted). A gully formed in the 1960s that grew progressively wider and deeper, and developed into a retrogressive thaw slump. In spring 2019, the slump diameter, which was determined using a UAV survey (Jongejans et al., 2021a), was about 890 m. Grow rates are fast with spatially and temporally varying headwall retreat rates of 7 up to 30 m y$^{-1}$ (Kunitsky et al., 2013; Günther et al., 2015; Vadakkedath et al., 2020). The ~55 m high headwall and the slopes of the slump provide access to stratigraphically discontinuous ancient permafrost deposits since the Middle Pleistocene (Murton et al., 2022). The headwall consists of six stratigraphical units from the bottom to the top: the Lower Ice Complex (Marine Isotope Stage (MIS) 16 or earlier), the Lower Sand Unit (sometime between MIS 16 and 6), the Woody Layer (MIS 5) which was present as lenses up to 3 m thick, the Upper Ice Complex (MIS 4-2) also called Yedoma, the Upper Sand Unit (MIS 3-2) and the Holocene Cover on top (MIS 1) (Ashastina et al., 2017; Murton et al., 2022). It should be noted that there are large hiatuses (marked by erosional surfaces below and above the Lower Sand) and dating uncertainties in the chronostratigraphy (Murton et al., 2022). While the ancient permafrost buried deep in the ground has survived multiple interglacials, the region has been subject to repeated permafrost thaw and sediment removal by thermo-erosional processes, amplified in recent decades.

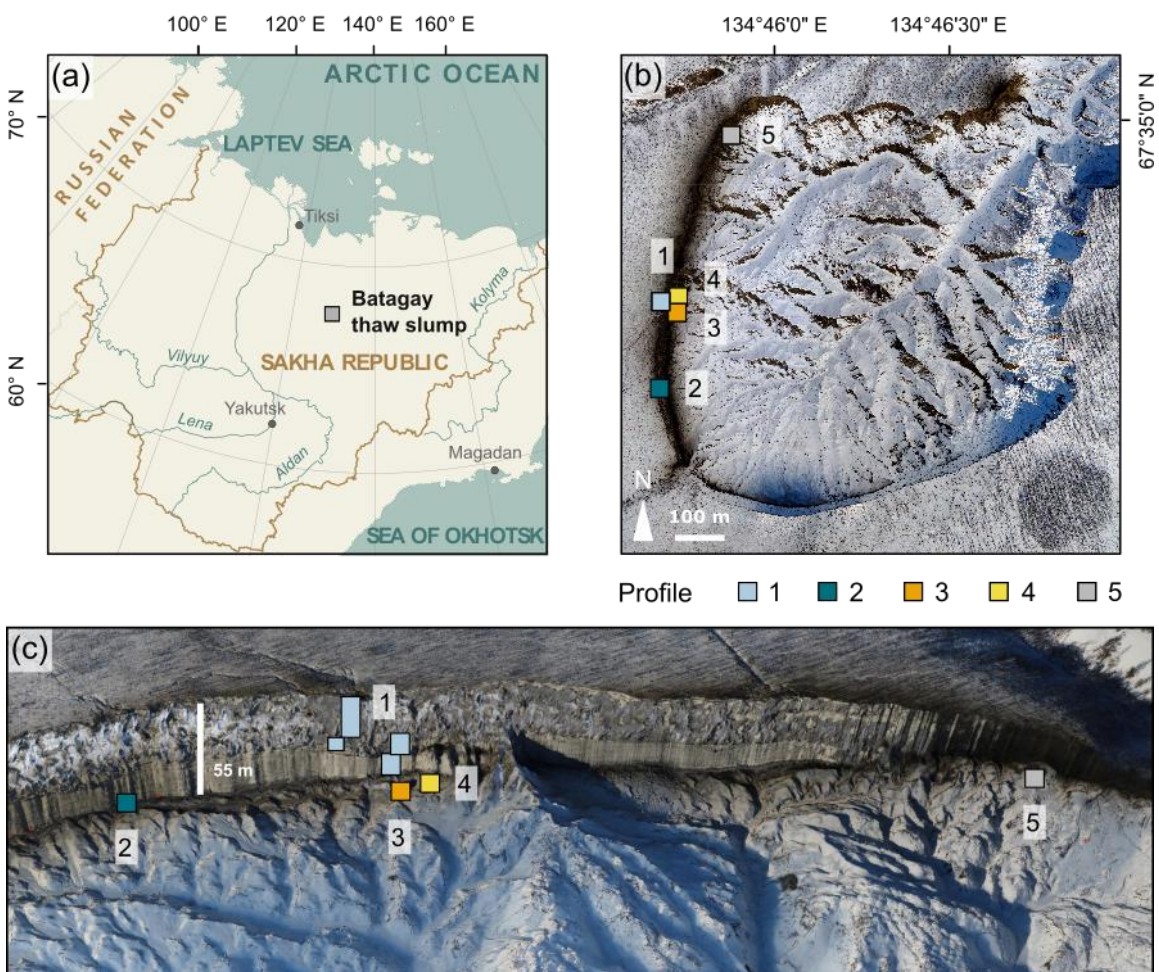

**Figure 1: Location of the Batagay thaw slump. (a) Overview of Yakutia (Republic Sakha) and location of the Batagay thaw slump (grey square), (b) aerial view of slump with sampling locations profile 1 to 5, and (c) front view of the east-facing headwall. Source (a): ESRI. Photos (b, c) from Spring Expedition to Batagay in 2019.**

## 3 Methods

### 3.1 Sample collection

The slump headwall was sampled during a spring expedition to Batagay in March and April 2019 (Figure 1b and c) (Jongejans et al., 2021a). The samples were taken by rappelling with a rope from the top of the slump headwall to each sample location and then using a hole saw (diameter: 57 mm, 40 mm deep) mounted on a hand-held power drill to sample small horizontal cores of frozen sediments exposed in the headwall. Sample depth is given in metres below the surface (m bs) (Figure S1). At each sampled depth, three cores were taken next to each other for biomarker, sedimentological and ancient DNA analyses. Sampling resolution was 0.5 m in the top 10 m and 1 m below. Due to the presence of large ice wedges, profile 1 consisted of

4 sub-profiles (Figure 1c, Figure A1). Using a hammer, axe and a chainsaw, more profiles were sampled at the lower part of the headwall from the slump bottom (profile 2; Figure A2), as well as at two large permafrost blocks at the slump bottom that have fallen from the headwall (profiles 3 and 4; Figure A3 and Figure A4, respectively), and at a baidzherakh (thermokarst mound) in the north of the slump (profile 5) (Figure 1b and c, Figure A5). All samples were stored in sterilised glass jars and kept frozen until laboratory analyses at AWI Potsdam. 30 samples (19 from profile 1, five from profile 2, and two each from profiles 3, 4 and 5) were selected for biomarker analysis. With these profiles, we covered five of the six stratigraphical units (all but the Upper Sand Unit which is not exposed in the central headwall). As we have no detailed sample depth information from the blocks and the baidzherakh, we report the results according to the respective stratigraphic units.

## 3.2 Laboratory analyses

The samples were freeze-dried and after homogenisation of the samples, the total carbon (TC), the total organic carbon (TOC; Vario TOC Cube Elemental Analyser), and the total nitrogen (TN) content were measured (Rapid MAX N exceed Elemental Analyser) and expressed in wt%.

Samples were treated for biomarker analysis as described by Jongejans et al. (2021b): after extraction of the OM (Dionex ASE 350) and removal of asphaltenes, four internal standards were added and the extracts were separated by medium pressure liquid chromatography (MPLC, Margot Köhnen-Willsch Chromatography, Jülich) into an aliphatic, aromatic and polar NSO (nitrogen, sulphur and oxygen-containing) fraction (for details see Radke et al., 1980). We selected 10 samples for further separation of the NSO fraction into an acid and neutral polar fraction using a KOH-impregnated silica gel column (Schulte et al., 2000). This sample selection was based on the biogeochemical parameters, as well as to cover the entire profile.

We measured alkanes, fatty acids (FAs) and alcohols using a TRACE 1310 Gas Chromatograph coupled to a TSQ 9000 Mass Spectrometer (Thermo Scientific), following the same method and settings as described in Jongejans et al. (2021b). Prior to the measurements, the fatty acid fraction was methylated using diazomethane and the alcohol fraction was trimethylsilylated using N-methyl-N-(trimethylsilyl)trifluoroacetamide (MSTFA). We quantified the compounds relative to the internal standards from full scan mass spectra (m/z 50-600 Da, 2.5 scans $s^{-1}$) using the software XCalibur.

We calculated indices from the $n$-alkane and $n$-FA concentrations (Table 1) to obtain insights into OM origin and preservation: the average chain length (ACL), the proxy for aquatic OM ($P_{aq}$), the carbon preference index (CPI), the ratio of $iso$- and $anteiso$-branched to long chain $n$-FAs (IA) and the higher-plants fatty acid (HPFA) index. The ACL can be used as an indicator of OM source, where long-chain $n$-alkanes (>25) are mostly produced by terrestrial higher plants (Poynter and Eglinton, 1990; Ficken et al., 1998; Zech et al., 2009). Variations of the ACL can be caused by different plant type material and climatic-induced changes of the environmental conditions. For example, different temperature and wetness conditions as well as the length of the vegetation period can influence the long chain $n$-alkane distribution (e.g., Sachse et al., 2006). The $P_{aq}$ shows the share of OM derived from aquatic plants, which are thought to contain relatively more $C_{23}$ and $C_{25}$ $n$-alkanes - compared to terrestrial plants which generally have longer chains (Ficken et al., 2000). In addition, $Sphagnum$ mosses are also dominated by $n$-$C_{23}$ and $n$-$C_{25}$. The CPI expresses the ratio of the odd over even $n$-alkane chains and decreases with OM decomposition

(Marzi et al., 1993). We calculated the IA using the *iso-* and *anteiso-*branched FAs $C_{15}$ and $C_{17}$ representing bacterial biomass relative to long chain *n*-FAs representing the terrestrial OM. This ratio is thought to reflect changes in the microbial abundance (and presumably activity) with respect to the terrestrial background biomass, where a higher ratio may correspond to microbial

membrane adaptation with respect to warmer environmental conditions (Rilfors et al., 1978; Stapel et al., 2016). Finally, the HPFA index was used to indicate the level of OM degradation: due to the presence of the polar carboxyl group, FAs are more vulnerable to biological and chemical degradation (Killops and Killops, 2013) compared to respective *n*-alkanes, leading to decreased HPFA values with decomposition.

**Table 1: Acronym and equations of calculated indices from *n*-alkane and *n*-fatty acid (*n*-FA) concentrations.**

| Index | Name | Equation |
|-------|------|----------|
| ACL | Average chain length of *n*-alkanes (Poynter and Eglinton, 1990) | $$ACL_{23-33} = \frac{\sum i \cdot C_i}{\sum C_i}$$ |
| $P_{aq}$ | Aquatic organic matter proxy, *n*-alkanes (Ficken et al., 2000) | $$P_{aq} = \frac{C_{23} + C_{25}}{C_{23} + C_{25} + C_{29} + C_{31}}$$ |
| CPI | Carbon preference index, *n*-alkanes (Marzi et al., 1993) | $$CPI = \frac{\sum odd\, C_{23-31} + \sum odd\, C_{25-33}}{2 \sum even\, C_{24-32}}$$ |
| IA | *Iso-* and *anteiso-* $C_{15}$ and $C_{17}$ FAs vs. long *n*-FAs | $$IA = \frac{iso + anteiso}{long\, n - FAs}$$ |
| HPFA | Higher-plant fatty-acid: *n*-FAs and *n*-alkanes (Strauss et al., 2015) | $$HPFA = \frac{\sum even\, n - FA\, C_{24-28}}{\sum even\, n - FA\, C_{24-28} + \sum odd\ n - alkane\, C_{27-31}}$$ |

## 4 Results

### 4.1 Detected biomolecules

We measured hydrocarbons in 30 samples. These compounds comprised short ($<C_{20}$) and long chain *n*-alkanes ($C_{20}$ to $C_{33}$), alkylcyclohexanes ($C_{17}$ to $C_{25}$), alkylcyclopentanes (even carbon numbered from $C_{18}$ to $C_{24}$), methylalkanes ($C_{19}$ to $C_{25}$), as

well as diethylalkanes and ethyl-methylalkanes ($C_{19}$ to $C_{25}$) (Figure S2). The concentrations of the branched (methylalkanes, diethylalkanes and ethyl-methylalkanes) and cyclic (alkylcyclohexanes and alkylcyclopentanes) alkanes strongly correlate with each other (correlation coefficient r: 0.97 to 0.99, *p*<0.01; Table S1).

Additionally, we measured FA concentrations of 10 samples. However, the acid fraction of the uppermost sample (0.2 m) likely contained a plastic contamination and coelutions with FAs prevented the quantification of FAs in this sample. Overall,

normal FAs (*n*-$C_{12}$ to $C_{34}$), *iso*-branched FAs (*iso*-$C_{14}$ to $C_{17}$), *anteiso*-branched FAs (*ai*-$C_{15}$ to $C_{17}$), saturated branched FAs (10-Me16, 10-Me17, 10-Me18, 12-Me18), monounsaturated FAs (16:1ω7, 16:1ω5, 18:1ω9, 18:1ω7, 19:1, 20:1ω9), a

polyunsaturated FA (18:2ω6-9), FAs with a cyclopropyl ring (cycl-17:0, cycl-19:0), hydroxyl FAs (22-OH, 24-OH) and phytanoic acid were found. In the neutral polar fraction, a homologous series of $n$-alcohols as well as sterols and triterpenoids were detected.

## 4.2 Lower Ice Complex

This lowermost exposed sediment sequence consisted mostly of sandy silt to silty sand. The Lower Ice Complex (profile 2: 53.1-52.0 m bs) contained partly truncated ice wedges and composite wedges. A reddish erosional layer containing gravel marked the top of the Lower Ice Complex. In places, a similar layer cuts through the Lower Ice Complex at an angle. Here, we found pool ice and wooden remains. The TOC (0.69-0.83 wt%) and the TN (0.10-0.11 wt%) were very low in this unit (Figure 2 and S1) and the C/N ratio ranged from 6.4 to 7.5. The concentrations of short $n$-alkanes (47-75 µg g$^{-1}$ TOC), long $n$-alkanes (213-405 µg g$^{-1}$ TOC) and branched and cyclic alkanes (46-161 µg g$^{-1}$ TOC) were also quite low in this unit. The ACL ranged between 28.5 and 29.2, and the $P_{aq}$ from 0.14 to 0.23. The CPI varied between 6.4 and 7.6. The main FAs were the $n$-FAs. The short and long chain $n$-FA concentrations were 163 and 610 µg g$^{-1}$ TOC in this unit, respectively (Figure 3). The IA index was very low (0.03) and the HPFA index was comparably high (0.67).

## 4.3 Lower Sand Unit

The Lower Sand Unit (profile 2: 51.5-51.0 m bs; profile 1: 49.4-38.4 m bs; one sample of profile 4) was characterised by narrow chimney-like composite ice-sand wedges. The TOC was higher (0.65 to 1.36 wt%) compared to the Lower Ice Complex, and the TN was comparably low (<0.10-0.13 wt%). The C/N ratio ranged from 7.6 to 10.8; it could only be calculated for the samples with a TOC and TN content above the detection limit. The alkane concentrations ranged between 13 and 145 µg g$^{-1}$ TOC for the short $n$-alkanes, 140 to 1329 µg g$^{-1}$ TOC for the long $n$-alkanes and between 41 and 553 µg g$^{-1}$ TOC for the branched and cyclic alkanes. The ACL and $P_{aq}$ ranged from 28.6 to 29.2 and 0.11 to 0.20, respectively. The CPI ranged between 7.2 and 11.5. The concentrations of short chain $n$-FAs spanned a large range from 130 to 432 µg g$^{-1}$ TOC and the long $n$-FAs ranged from 214 to 447 µg g$^{-1}$ TOC. The IA was on the low end (0.04 to 0.08) and the HPFA index was between 0.18 and 0.58.

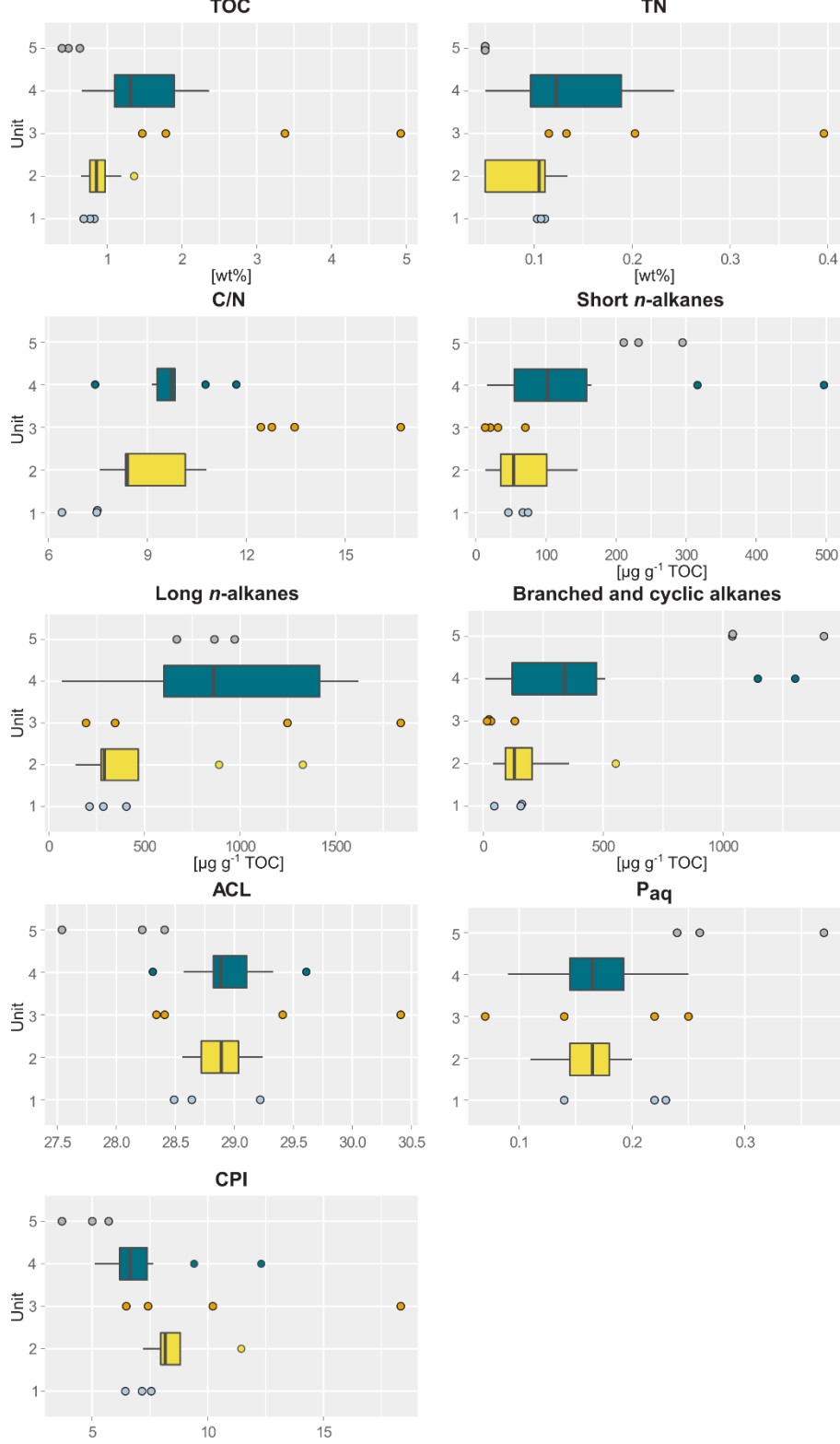

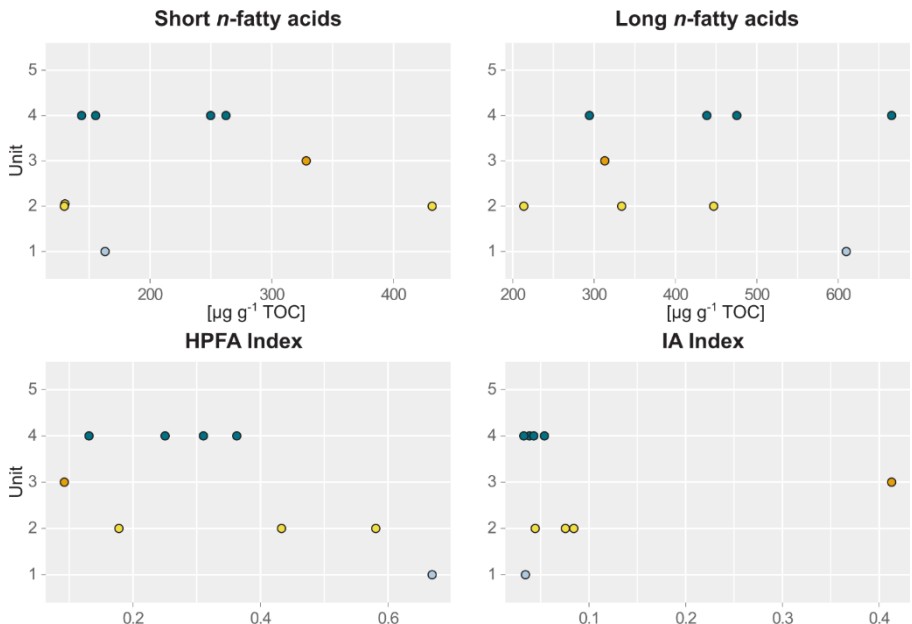

**Figure 2: Boxplots of biogeochemical and alkane parameters of sediment samples taken from five different profile units; 1. Lower Ice Complex (n=3), 2. Lower Sand Unit (n=12), 3. Woody Layer (n=4), 4. Upper Ice Complex (n=8), 5. Holocene Cover (n=3) of the main headwall of the Batagay megaslump. From left to right and top to bottom: total organic carbon content (TOC), total nitrogen content (TN), carbon to nitrogen ratio (C/N), short chain *n*-alkane concentration, long chain *n*-alkane concentration, concentration of branched, cyclic and methylalkanes combined, average chain length of *n*-alkanes (ACL), aquatic organic matter proxy $P_{aq}$, and carbon preference index (CPI). For units with <5 samples, beeswarm plots are shown instead of boxplots.**

**Figure 3: Beeswarm plots of fatty acid concentrations of sediment samples of main headwall. From left to right and top to bottom: short chain *n*-fatty acid (*n*-FA) concentration, long chain *n*-FA concentration, ratio of *iso-* and *anteiso*-FA $C_{15}$ and $C_{17}$ vs long FAs (IA index) and higher-plants fatty acid (HPFA) index. Units: 1. Lower Ice Complex (n=1), 2. Lower Sand Unit (n=3), 3. Woody Layer (n=1), 4. Upper Ice Complex (n=4).**

### 4.4 Woody Layer

The Woody Layer (profile 1: 33.5-31.7 m bs; one sample each of profiles 3 and 4) was present in lenses up to 3 m thick. This debris layer was abundant in organic remains, peat lenses, roots, and wood. The TOC (1.47 to 4.93 wt%) and TN (0.12 to 0.40 wt%), as well as the C/N ratio (12.4 to 16.7) were highest in this unit. Here, the short *n*-alkanes, branched and cyclic alkanes were scarce (13-71 and 16-132 µg g$^{-1}$ TOC, respectively; Figure S2), but the long chain *n*-alkanes covered a large range (194-1841 µg g$^{-1}$ TOC). The ACL (28.3-30.4) had its maximum in this unit and the $P_{aq}$ (0.07-0.25) its minimum (both at 31.7 m bs in profile 1). The CPI was moderate to high (6.5 to 18.3). In this unit, we analysed the neutral fraction of one sample: the sample at 31.7 m bs from profile 1. In this sample, the *iso-* and *anteiso*-FAs (as well as the unsaturated FAs) were most abundant (Figure S3) and therefore, the IA value was the highest (0.41). The FA concentrations were 328 µg g$^{-1}$ TOC for the short and 313 µg g$^{-1}$ TOC for the long chain *n*-FAs. The HPFA index was very low (0.09). Furthermore, we found many different sterols and triterpenoids in this sample (Table 2). The gas chromatogram and molecular structures can be found in

the supplements (Figures S4 and S5). In the samples from the other units (n=9), we found only the sterols campesterol and β-sitosterol.

## 4.5 Upper Ice Complex – Yedoma

The Upper Ice Complex (profile 1: 30.7-4.2 m bs; one sample of profile 3; profile 5) contained large (up to a few meters wide) syngenetic ice wedges. The TOC (0.66-2.36 wt%) and TN (<0.10-0.24 wt%) contents were moderately high compared to the other units. The C/N values (7.4-11.7) were very similar to those of the Lower Sand Unit. Alkane concentrations spanned a wide range in the Upper Ice Complex: 16-497 µg g$^{-1}$ TOC for the short chain $n$-alkanes, 68-1620 µg g$^{-1}$ TOC for the long $n$-alkanes and 8-1302 µg g$^{-1}$ TOC for the branched and cyclic alkanes. The ACL and P$_{aq}$ spanned quite a wide range (28.6 to 29.2 and 0.11 to 0.20, respectively). The CPI was low to moderate in this unit (5.11 to 12.3). The $n$-FA concentrations were also quite variable with the short chain $n$-FAs ranging between 144 and 262 µg g$^{-1}$ TOC and the long chain $n$-FAs between 294 and 666 µg g$^{-1}$ TOC. The IA index was very low (0.03 to 0.05) and the HPFA index low to medium (0.13 to 0.36).

## 4.6 Holocene Cover

The Holocene Cover Unit (profile 1: 2.0-0.2 m bs) seemed quite organic-rich and contained a variety of cryostructures (e.g. massive, porphyric, basal, belt-like and layered). Nevertheless, the TOC (0.39 to 0.63 wt%) and TN (<0.10 wt%) values were very low. Due to the TN values below the detection limit, we could not calculate the C/N values of this unit. Especially the branched and cyclic alkanes were very abundant (790-1422 µg g$^{-1}$ TOC), whereas the short (211-295 µg g$^{-1}$ TOC) and long chain $n$-alkanes (669-972 µg g$^{-1}$ TOC) were moderately high. The ACL (27.5-28.4) was lowest in this unit and the P$_{aq}$ the highest (0.24-0.37) in all profiles. The CPI was also lowest and ranged from 3.7 to 5.7.

**Table 2: Identified sterols and triterpenoids in sediments in the Woody Layer at 31.7 m bs (profile 1) of the Batagay megaslump. RT: gas chromatography-mass spectrometry (GC-MS) retention time; $M^+$: molecular ion, a: as trimethylsilyl derivative. Bold characteristic fragments represent the most intense fragment (base peak) in the mass spectra. The GC-MS chromatogram and the chemical structures of the compounds are shown in the supplements (Figure S4 and S5, respectively).**

| No. | Trivial name | Full name | RT [min] | $M^+$ [m/z] | Characteristic fragments [m/z] |
|---|---|---|---|---|---|
| 1 | Cholesterol | Cholest-5-en-3β-ol | 89.55 | 458 [a] | 368, 329, 159, 145, **129** |
| 2 | Cholestanol | 5α(H)-Cholestan-3β-ol | 89.87 | 460 [a] | 445, 403, 370, 355, 305, 215, **75** |
| 3 | Brassicasterol | 24-Methylcholesta-5,22-dien-3β-ol | 90.50 | 470 [a] | 380, 341, 255, 129, **69** |
| 4 | Campesterol | 24-Methylcholest-5-en-3β-ol | 92.15 | 472 [a] | 457, 382, 367, 343, 255, **129** |
| 5 | Stigmasterol | 24-Ethylcholesta-5,22-dien-3β-ol | 92.71 | 484 [a] | 394, 255, 129, **83** |
| 6 | β-Sitosterol | 24-Ethylcholest-5-en-3β-ol | 94.60 | 486 [a] | 471, 396, 381, 357, 255, **129** |
| 7 | Stigmastanol | 24-Ethylcholestan-3β-ol | 94.96 | 488 [a] | 473, 398, 383, 305, **215** |
| 8 | β-Amyrin | Olean-12-en-3β-ol | 95.77 | 498 [a] | **218**, 203, 189 |
| 9 | Stigmast-7-en-3β-ol | 24-Ethylcholest-7-en-3β-ol | 96.55 | 486 [a] | 471, 381, **255**, 229, 213, 75 |
| 10 | α-Amyrin | Urs-12-en-3β-ol | 97.29 | 498 [a] | **218**, 203, 189 |
| 11 | Oleanenone | Olean-12-en-3-one | 97.77 | 424 | 409, **218**, 203, 189 |
| 12 | β-Saccharostenone | 24-Ethylcholesta-3,5-dien-7-one | 98.41 | 410 | 395, 269, 187, **174**, 161, 159 |
| 13 | Sitostenone | 24-Ethylcholest-4-en-3-one | 99.73 | 412 | 370, 289, 229, **124** |
| 14 | Lupeol | Lup-20(29)-en-3β-ol | 100.37 | 498 [a] | 483, 393, 369, **189** |

## 5 Discussion

Variations in the TOC contents and fossil biomolecule concentrations along the sedimentary succession provide insights into quantitative differences of the buried OM deposited over time. These differences are mainly caused by changes in the depositional regime (e.g. water availability, temperature, accumulation rates), the associated bioproductivity (autochthonous signal) and transport processes of the OM (allochthonous signal) following different climatic periods (e.g., glacial and interglacial periods). Additionally, qualitative variations of the fossil biomolecules can give insight into different OM sources such as the biomarker indices ACL and $P_{aq}$. Indicative biomarkers are a useful tool in these old sediments, as they are generally very well preserved in sediments, even on geological timescales, compared to for example sugars, proteins and DNA.

**Biogeochemical legacy of glacial periods**

In the Batagay dataset, we found generally only minor variations in the biogeochemical and biomarker parameters for the Lower Ice Complex, Lower Sand Unit and Upper Ice Complex. This suggests that the OM signal representing permafrost

deposits since about 650 ka is qualitatively similar, suggesting that vegetation patterns might have been similar over time in glacial periods. These observations fit well with the palaeo-vegetation records of Ashastina et al. (2018). They found that meadow-steppe vegetation persisted throughout most of the reconstructed period (i.e. Lower Sand Unit and Upper Ice Complex) and argued that fossil plant macro-remains mirror mostly changes in the relative abundance of plant communities rather than complete changes in plant species compositions over time (Ashastina et al., 2018). Such relative quantitative

variations in the vegetation might be responsible for the observed variability in individual biomolecule markers (e.g. *n*-alkanes and FAs). For the MIS 3 and MIS 2 deposits, Courtin et al. (accepted) confirmed the open steppe-tundra landscape by sedimentary DNA analyses; they revealed that herb communities dominated the glacial vegetation and they found traces of megaherbivores corresponding to this landscape.

The generally higher ACL (>28) and lower $P_{aq}$ in these units indicate a more higher-plant and less aquatic or mossy character

of the OM in these deposits. This corroborates the strong continentality and dry conditions, especially during the cold stages, as found by isotopic and palaeo-ecological analyses (Ashastina et al., 2018; Opel et al., 2019). The relatively low IA index presumably points to lower microbial activity during the glacial periods.

Cryostratigraphic observations and isotopic findings suggest that the Lower Ice Complex sediments might have been deposited under relatively wet conditions providing enough snowmelt water to form huge ice wedges (Opel et al., 2019). These findings

suggest that these sediments were deposited during a glacial period. In contrast, shotgun DNA analyses from sediments taken in 2017 from the upper part of the Lower Ice Complex just below the erosional surface (sample B17-D3) point to an interglacial origin of the deposited OM (Courtin et al., accepted). Courtin et al. (accepted) suggested that the environment was characterised by forested vegetation, but that there were also more open, herb-dominated areas with large herbivores. Pollen findings (A. Andreev, unpublished data) of the same samples from the Lower Ice Complex at its transition into the above-lying erosional

surface point to woodland and steppe vegetation, characteristic of an interglacial period that might have induced thermo-erosion and permafrost thaw that partly degraded the Lower Ice Complex from above. In the sediments above the erosional surface, in the Lower Sand Unit (sample B17-D5), Courtin et al. (accepted) detected small mammals and forest-specific insect families supporting dense forest vegetation. Furthermore, they found signs for strong microbial activity related to soil decomposition such as members of the fungus *Pseudogymnoascus*, which are related to decaying roots or plants, and aerobic

bacteria (Nocardioidaceae and *Clostridia*) which are considered to be consumers of OM. In contrast to this transition layer, the samples of the underlying Lower Ice Complex taken in 2019 cover the entire exposed sequence and our biochemical and biomarker results do not differ for the Lower Ice Complex, the Lower Sand and the Upper Ice Complex. Therefore, we assume that all three sequences formed during glacial periods. Moreover, we found relatively low values for the IA index in the Lower Ice Complex deposits, suggesting low microbial activity. Possibly the samples from the Lower Ice Complex (2017 and 2019)

represent a transition from a glacial into an interglacial period of which the latter is represented in the erosional surface topping the Lower Ice Complex. Apart from the erosional surface above the Lower Ice Complex (Figure 4 from Opel et al., 2019), there were signs of erosion events within the Lower Ice Complex as indicated by pockets of wooden remains (Jongejans et al., 2021a). In any case, the complicated permafrost formation and degradation history might also explain the mixed signal in the

OM: the C/N ratio and HPFA index show opposite results for the Lower Ice Complex. The high HPFA index might be influenced by the high long $n$-FA concentration. The low C/N could point to the deposition of older transported OM. The CPI was strongly correlated with the ACL (r: 0.74, $p<0.01$) and $P_{aq}$ (r: -0.70, $p<0.01$) across all units (Table S2). This suggests that the CPI is highly influenced by the OM source and therefore, its use as an OM quality indicator might be restricted. However, general CPI values above 5 might indicate that the OM is still of relatively good quality. A deeper insight on the quality might be provided by the FA concentrations as they are indicators for more labile biomolecules. The FA data show quite variable values within the individual glacial periods (Figure 3). In addition to a mixed OM source, this might also indicate a heterogeneous level of OM decomposition, which is supported also by variable HPFA values. Thus, the data point to an overall variable OM quality in the glacial deposits.

The occurrence of narrow composite sand-ice wedges in the Lower Sand Unit compared to the large ice-wedges in both Ice Complex units suggests very high accumulation rates in the Lower Sand Unit. Furthermore, there was likely more snowmelt water available during the Ice Complex formation that allowed the formation of huge ice wedges as present in the Lower and the Upper Ice Complex units. Nevertheless, these changes in available winter moisture are not reflected in the biomarker record of the e.g., ACL and $P_{aq}$ values.

## 5.2 Biogeochemical legacy of interglacial periods

In contrast, the Woody Layer and the Holocene Cover differ in biogeochemical and biomarker parameters from the other stratigraphic units. Compared to the glacial units, we found distinct differences in the $n$-alkane and FA distribution for the Holocene Cover and the Woody Layer, but also some specific biomarkers in these sediments such as branched and cyclic alkanes, stenols, stanols and pentacyclic triterpenoids. We discuss the characteristics of the OM in these sediments, and the sources and implications of these compounds in the Woody Layer and the Holocene Cover below.

The Woody Layer samples show a wide variability among all determined biogeochemical and biomolecular parameters indicating a layer of high inhomogeneity. Ashastina et al. (2017) found high TOC and C/N values, as well as low $\delta^{13}$C values for the Woody Layer. Similarly, we found variable but overall higher TOC contents in these sediments pointing to a high OM accumulation in this layer, and compared to the other units, a higher C/N and $ACL_{23-33}$ suggesting a strong higher-plant contribution in the deposited OM. However, a variable input of aquatic or mossy organic biomass is indicated by the $P_{aq}$ index. The higher OM accumulation could result from higher productivity as typical for warmer conditions during interglacial periods. However, the fact that the Woody Layer marks a disconformity related to massive permafrost degradation and erosion suggests that the OM can also stem from remobilisation of older material, redistribution, and accumulation in erosional forms.

The sediments of the Woody Layer had a distinctly different $n$-alkane and FA distribution compared to the other studied sediment units. The Woody Layer almost completely lacked the short $n$-alkanes, and branched and cyclic alkanes, and the high ACL and low $P_{aq}$ suggest drier conditions (Ficken et al., 1998, 2000). Apart from the distinct $n$-alkane and FA distribution, the sediments from the Woody Layer (sample at 31.7 m in profile 1) also contained specific stenols, stanols and pentacyclic triterpenoids (Table 2). While it is thought that $C_{27}$ and $C_{28}$ sterols dominate in algae and zooplankton, $C_{29}$ sterols are generally

abundant in vascular plants (Volkman, 1986). Furthermore, many of the compounds identified in the Batagay sediments were found to be typical for higher land plants: campesterol, stigmasterol, β-sitosterol, stigmastanol, β-amyrin, α-amyrin, oleanenone and lupeol (Brassell et al., 1983; Peters et al., 2005; Killops and Killops, 2013). The presence of these markers point to a strong terrestrial signal of OM, which is partly corroborated by the high ACL and lower $P_{aq}$ values in the Woody Layer sediments. These findings match those of Ashastina et al. (2017) who found no aquatic or wetland plants for this unit, but only terrestrial plant remains.

The Woody Layer accumulated in an erosional gully, which is indicated by the presence of organic-rich lenses and abundant trash-wood in the headwall. Similar "forest beds" that were associated with the Last Interglacial were found in non-glaciated Yukon and Alaska (Hamilton and Brigham-Grette, 1991; Reyes et al., 2010). In the Woody Layer, a mixture of different autochthonous and allochthonous organic biomass was transported and accumulated. Thermo-erosional processes such as the formation of gullies (the combined mechanical and thermal action of moving water) (van Everdingen, 2005), is associated with running or standing water that can transport sediments and organic remains. However, aquatic markers are only present in minor abundance but might be represented by short chain FAs and sterols such as brassicasterol (Killops and Killops, 2013). In addition, Ashastina et al. (2018) reconstructed dry conditions during the Last Interglacial with a herb-rich light coniferous taiga and a pronounced plant litter cover. They argued that this could be related to the low ice content of the underlying Lower Sand Unit, providing little meltwater from thawing permafrost. Furthermore, they found that plant and insect species composition pointed to frequent fire disturbances in the Last Interglacial. The high abundance of *iso-* and *anteiso* FAs (IA index) as well as high amounts of branched and unsaturated short chain FAs (Figure S3) suggests increased microbial activity for this interval (Stapel et al., 2016). Together with the very low HPFA index, this indicates an increased level of microbial transformation of the OM, and thus a lower quality of the OM in the Woody layer.

In the Holocene Cover sediments, the relatively low ACL and high $P_{aq}$ values suggest an increasing amount of aquatic plants formed under wet conditions or mosses (Ficken et al., 1998, 2000). In the sediments of the Holocene Cover and some samples from the Upper Ice Complex, the short *n*-alkanes were abundant. Especially, in these sediments, we found the presence of branched and cyclic alkanes. The branched alkanes, among which are the diethylalkanes and the ethyl-methylalkanes, have one or two quaternary carbon atoms (branched aliphatic alkanes with a quaternary substituted carbon atom: BAQCs). Kenig et al. (2005) argued that the BAQCs are widespread in sediments and sedimentary rocks due to their low biodegradability, but were not identified often or misidentified before. The source of these, as well as of the cyclic alkanes (alkylcyclohexanes and alkylcyclopentanes) and methylalkanes, has been a topic of debate (e.g., Shiea et al., 1990; Greenwood et al., 2004; Kenig et al., 2005). The strong positive correlation (r>0.97) between the concentrations of the BAQCs and cyclic alkanes suggests similar sources for these compounds. Previous studies also found the co-occurrence of these compounds (e.g., Ogihara and Ishiwatari, 1998; Kenig et al., 2005). Several studies proposed a microbial origin, such as cyanobacteria (Shiea et al., 1990), non-photosynthetic sulfidic oxidising bacteria (Kenig et al., 2003), thermophilic acidophilic bacteria (Ogihara and Ishiwatari, 1998) or microbes exploiting redox gradients or involved in either the sulphur or nitrogen cycle (Greenwood et al., 2004). Zhang et al. (2014) suggested that the long-chain cyclic alkanes could be produced by the reduction of FAs. Cyanobacteria

could have been present in polygonal ponds, running water or even in liquid pore water. However, we did not find a correlation with concentrations of certain FAs that are major components produced by cyanobacteria such as 16:0, 16:1ω7 and 18:1ω9 (Piorreck et al., 1984). Nevertheless, these FAs are not very specific and thus can be a signal of different sources preventing a direct correlation to the alkylated and cyclic alkanes. A plastic contamination was also proposed as the source of BAQCs by

Brocks (2008), but we would expect that previous studies where sediment samples were prepared in a similar way would have found these compounds as well (e.g., Strauss et al., 2015; Jongejans et al., 2018, 2020, 2021b). Further, a petroleum contamination can be ruled out as we did not find corresponding oil-related geothermally transformed compounds such as hopanes and steranes. Further research is needed to be able to reduce the amount of possible sources. Generally, we assume a microbial origin for the branched and cyclic alkanes. This is corroborated by the strong positive correlation between the

branched and cyclic alkanes, and the short $n$-alkanes (r: 0.90, $p<0.01$). Also, even though the correlation was not significant when looking at the complete sample set, higher concentrations of branched and cyclic alkanes did match lower ACL and higher $P_{aq}$ values. These findings suggest that these alkanes are also produced under relatively warmer and wetter conditions which fits the Holocene origin of these samples very well. The low TOC contents and lowest CPI values suggest a higher degradation level and thus lower quality for the Holocene OM. Our findings point to drier conditions during the last interglacial

compared to the Holocene, as well as more bioproductivity and microbial degradation, indicating higher temperatures. This fits nicely to the findings of Kienast et al. (Kienast et al., 2008).

Altogether, it would be expected that there is a distinct difference between the Upper Ice Complex and the Holocene Cover. Still, it is likely that the uppermost part of the Upper Ice Complex was degraded during the Holocene. This might have led to a rather gradual transition of the biogeochemical and biomarker parameters within the Holocene Cover sediments and into the

Upper Ice Complex.

**5.3 Modern organic matter mobilisation in the Batagay megaslump**

Using satellite imagery, Vadakkedath et al. (2020) analysed the expansion of the thaw slump for the past three decades (1991-2018) and found increasing expansion rates over time with a mean of 2.6 ha y$^{-1}$. This means that an enormous amount of sediments and OM is mobilised every year. Following the thaw of the ice-rich sediments (especially of the Lower and Upper

Ice Complex units), the mobilised material can be transported by the meltwater rapidly downslope through a gully network into the Batagay River and further into the Yana River. The OM in these sediments can be decomposed by microbes upon thaw, leading to greenhouse gas emission from the sediments directly (Vonk et al., 2013) or from rivers.

Intense permafrost thaw occurred during interglacials and we found stratigraphic discordances above the Lower Ice Complex, the Lower Sand Unit and the Lower Ice Complex. Nevertheless, the presence of large ice wedges in the Lower and the Upper

Ice Complex and composite wedges in the Lower Sand Unit, shows that the sediments that are still exposed in the Batagay megaslump were affected only in their upper parts and remained largely undisturbed. Hence, OM decomposition was presumably limited. Previous studies showed the high lability of OM in permafrost and especially in the MIS 4-2 Yedoma Ice Complex sediments (Vonk et al., 2013; Jongejans et al., 2021b). Although the biomarkers indicate variable OM quality for the

different sedimentary intervals, we expect that still a large amount of biodegradable OM is mobilised from the Batagay thaw slump every thawing season. From the glacial and Holocene deposits, mostly mineral OM is mobilised, whereas from the Woody Layer, well-preserved OM including wooden remains and detritus are mobilised which can be readily decomposed upon thaw. The increased formation of retrogressive thaw slumps that has been observed over the past decades in many arctic regions (e.g., Lacelle et al., 2010; Lewkowicz and Way, 2019), is likely to continue with ongoing climate warming, and the mobilisation of large amounts of previously frozen sediments and OM likely will lead to higher GHG release from thawing permafrost (Bröder et al., 2021; Mann et al., 2022; Yao et al., 2021).

Multiple studies pointed to accelerating rapid degradation of ice-rich permafrost landscapes by thaw slumping, including regions with buried glacial ice but also regions with large syngenetic Yedoma ice wedges (Lantz and Kokelj, 2008; Lacelle et al., 2010; Kokelj et al., 2017; Lewkowicz and Way, 2019; Runge et al., 2022). In their study of thaw slumps in northwestern Canada, Lacelle et al. (2015) found 189 active slumps of which 10 exceeded 20 ha. However, recent remote sensing work on thaw slumps (e.g., Kokelj et al., 2015; Runge et al., 2022) suggested that mega slumps (up to 52 ha or larger) are rather rare so far. Therefore, at this point the Batagay thaw slump is very unique in its size and the largest feature as far as we know. As the initial disturbance of the Batagay megaslump is possibly anthropogenic, it represents an outstanding example of rapid permafrost thaw that is promoted, but was not originally caused by arctic warming.

## 6 Conclusions

Biogeochemical analyses provide valuable information on palaeo environments. Here, for the first time ancient permafrost that has formed since about 650 ka ago in NE Siberia was studied for carbon and nitrogen contents, and lipid biomarker characteristics. Our findings show that there was no substantial vegetation change of prevailing meadow steppe over large glacial periods during MIS 16, sometime between MIS 16 and MIS 6, and MIS 4-2, which are represented in the exposed strata of the Batagay megaslump by the Lower Ice Complex, Lower Sand Unit and the Upper Ice Complex, respectively. The interglacial Woody Layer (MIS 5), a layer of eroded and accumulated material, showed a high content of higher-plant OM and strong microbial decomposition. In the Holocene Cover, we found relatively wet depositional conditions. For the interglacial periods, the biomolecule inventory indicates a higher microbial OM transformation and thus a decreased OM quality. In contrast, in the glacial periods a variable but overall higher OM quality is suggested by the biomolecules compared to the interglacial periods. Thus, microbial decomposition was likely limited during the glacial periods. Therefore, a substantial amount of less decomposed OM is mobilised in the Batagay thaw slump every year, in particular since the thaw slump process allows access to deeply buried OM. Our biomarker analyses of ancient permafrost sediments contribute to a better understanding of how OM is incorporated and preserved in permafrost deposits during glacial and interglacial periods. Furthermore, it helps to improve our comprehension of possible consequences resulting from future permafrost thaw and OM mobilisation.

**Acknowledgements**

We would like to thank Dmitry Ukhin for his support in sample collection, and Ilya Kozhenikov and Stepan Vasiliev from the Melnikov Permafrost Institute SB RAN Yakutsk for their logistical support during the field work. We thank Justin Lindemann, Angélique Opitz, Jonas Sernau (AWI), and Anke Sobotta (GFZ) for their assistance in the laboratory. We thank AWI logistics for their help in the field work logistics.

**Funding information**

LJ was supported by a PhD Scholarship from the German Federal Environmental Foundation (DBU). TO acknowledges funding by the Leverhulme Trust Research Project Grant RPG-2020-334. AK was supported by the Lomonosov Moscow State University program "The cryosphere evolution under climate change and anthropogenic impact" (#121051100164-0). AWI provided baseline funding for expedition logistics and sample processing.

**Conflict of interest**

The authors declare that there is no conflict of interest.

**Author contributions**

LJ and JS were responsible for the conceptualisation of the research. Fieldwork was carried out by LJ, TO, JC, HM, SW, AK, AS and IS. Data acquisition and analysis was done by LJ, KM, CK and JS. LJ wrote the original draft; all authors contributed to the review and editing of the manuscript.

**Data availability statement**

Upon publication, all data presented in this study will be freely accessible in the PANGAEA Data Repository.

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

**Appendix A**

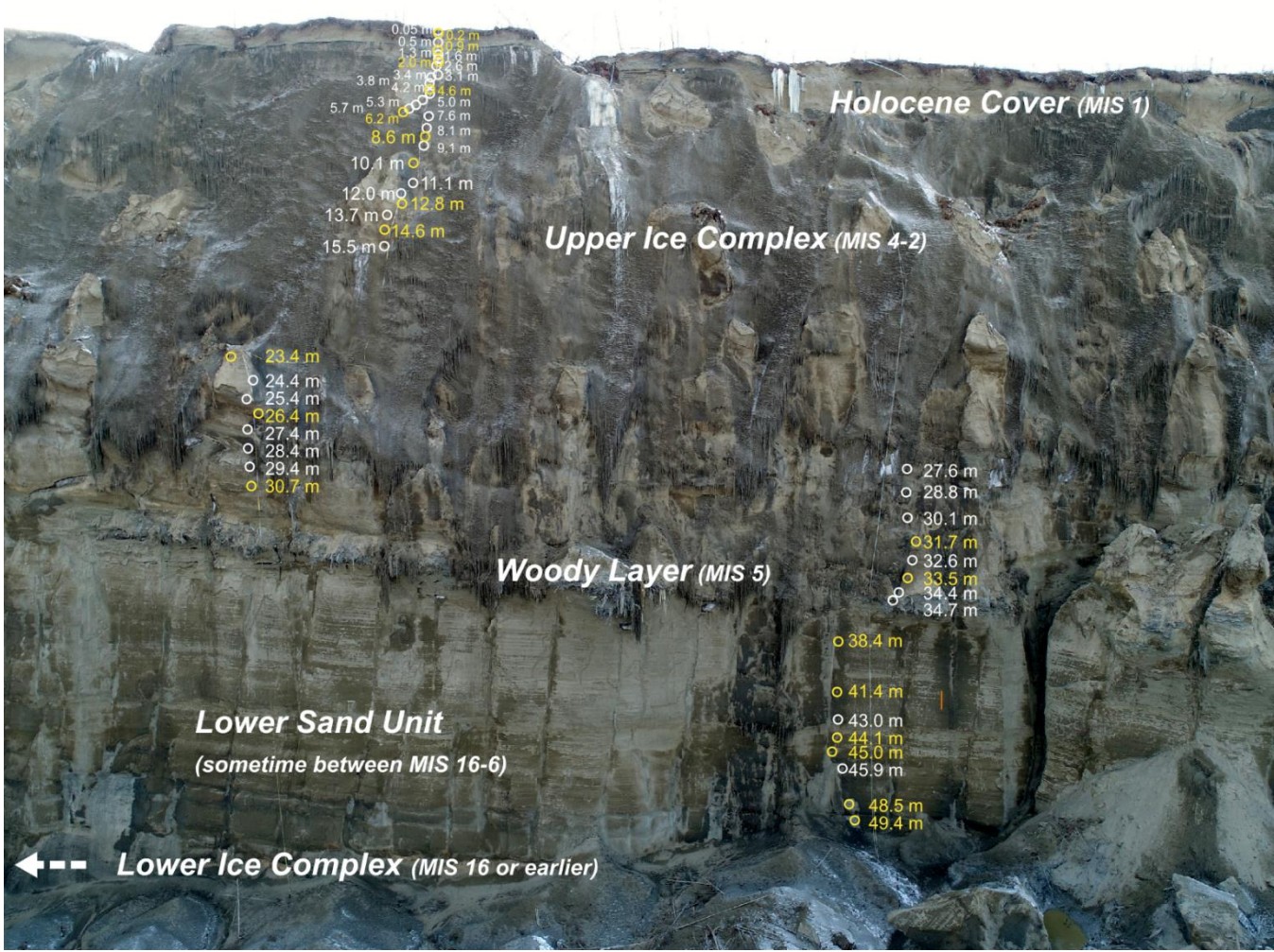

**Figure A1: Sediment profile B19-P1 (67.58004° N, 134.76130° E) at the west wall of the Batagay thaw slump. Position of sediment samples for biomarker analyses (yellow circles) and other samples (white circles) indicated. Photos in Figure 1-5 from Spring**
**Expedition to Batagay in 2019. Stratigraphical units: Lower Ice Complex, Lower Sand Unit, Woody Layer, Upper Ice Complex and Holocene Cover. MIS: Marine Isotope Stage. Depth in m below surface. The headwall is ~55 m high.**

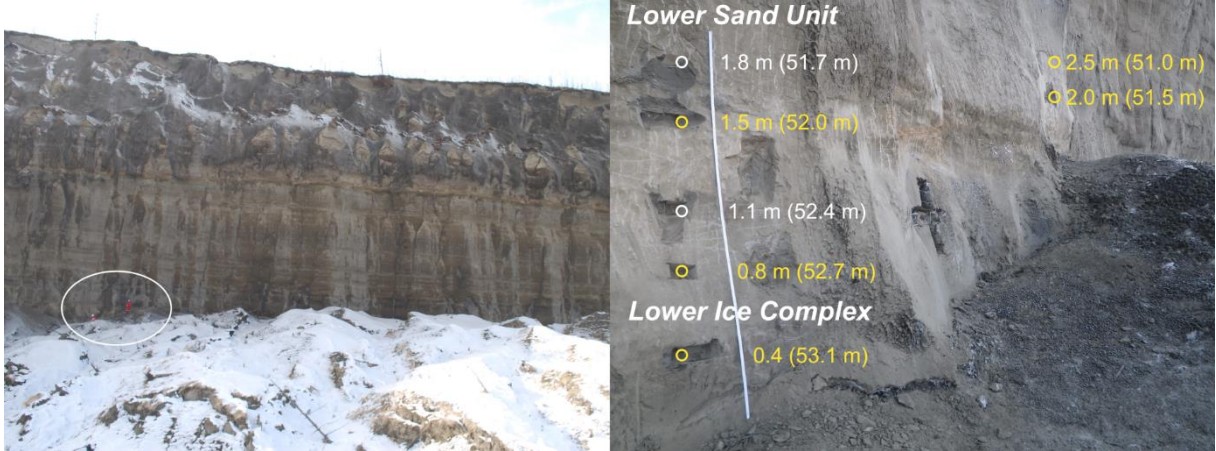

**Figure A2: Sediment profile B19-02 (67.57845° N, 134.76131° E). Left: position of profile on the west wall (white circle). Right: headwall profile with position of sediment samples for biomarker analyses (yellow circles) and other samples (white circles) indicated. Depth in m above slump bottom and approximate depth in m below surface respective to B19-P1 in brackets. Figures A2 to A5: adapted from Jongejans et al. (2021a), stratigraphic units as in Figure A1.**

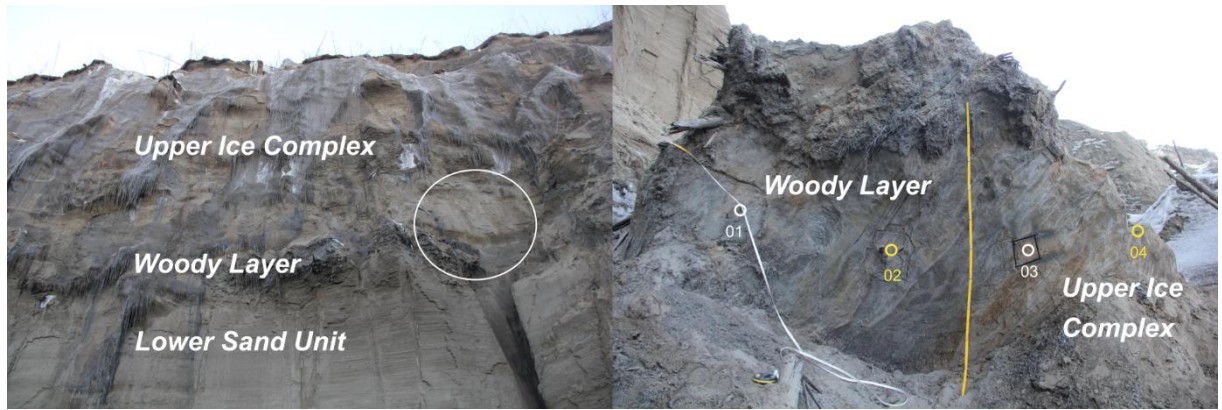

**Figure A3: Sediment profile B19-03 (67.58004° N, 134.76130° E). Left: approximate position where the block has fallen from (white circle). Right: block in the slump floor with position of sediment samples for biomarker analyses (yellow circles) and other samples (white circles) indicated.**

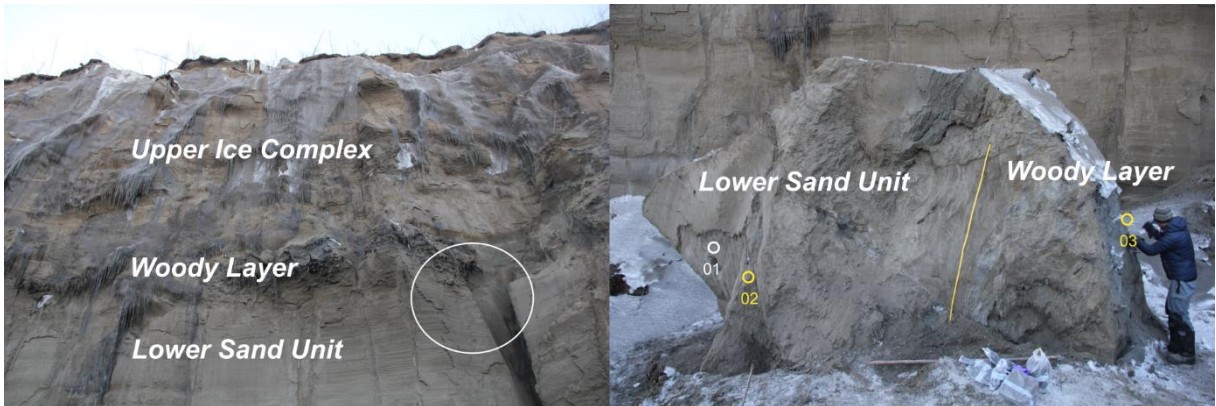

**Figure A4: Sediment profile B19-04 (67.58004° N, 134.76130° E). Left: approximate position where the block has fallen from (white circle). Right: block in the slump floor with position of sediment samples for biomarker analyses (yellow circles) and other samples (white circles) indicated.**

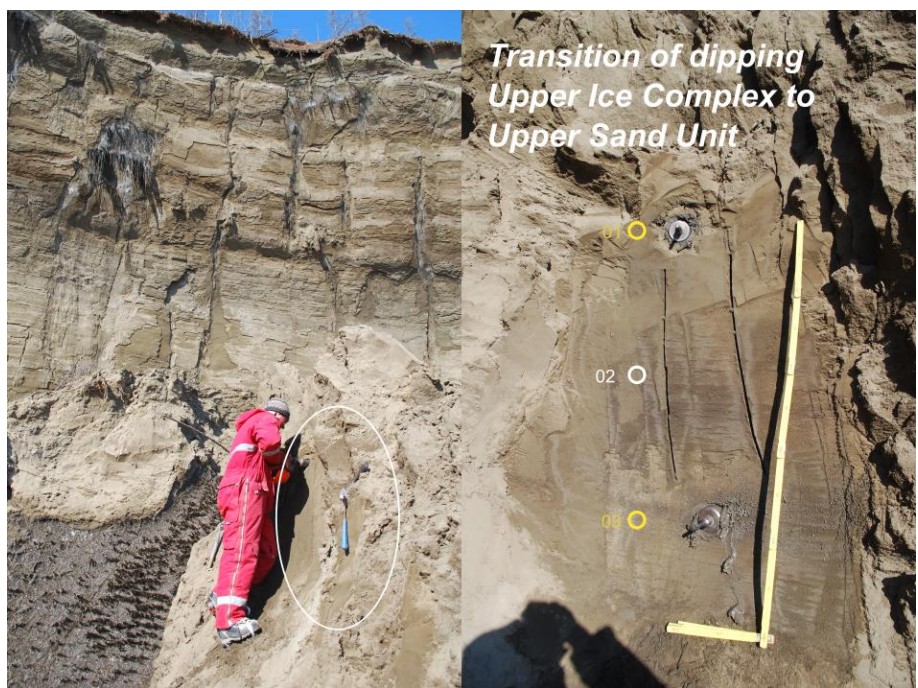

**Figure A5: Sediment profile B19-05 (67.58300° N, 134.76437° E). Left: position of profile on the baidzerakh at the slump floor (white circle). Right: baidzerakh profile with position of sediment samples for biomarker analyses (yellow circles) and other samples (white circles) indicated.**