# Peer review of "Molecular biomarkers in Batagay megaslump permafrost deposits reveal clear differences in organic matter preservation between glacial and interglacial periods"

_The Cryosphere, 2022_

## Author Comment (AC1)

**Reviewer 1**

Jongejans et al. present a study that uses bulk and organic geochemical measurements to investigate the organic matter stored in the ~55 m headwall of the Batagay slump. This is quite intriguing as this slump is a unique insight into a long history of permafrost accumulation (and degradation) that may be useful for both understanding the evolution of permafrost landscapes as well as predicting future impacts of carbon stored in permafrost sediments as the environment is transformed by anthropogenic climate change. Overall, I find their manuscript to be well-written and quite detailed, but not overly long. I deeply appreciate the technical detail presented in the biomarker work (Table 2 is a delight) that is sometimes overlooked or omitted by organic geochemists.

Thank you for your constructive feedback to our manuscript. We responded to all comments below. Note: the line numbers refer to the preprint version

I have relatively few questions and comments as this paper seems to fit nicely into the recent series of papers involving the ongoing investigation of the Batagay slump. Here are a few worth considering:

- The authors note that the interglacial units appear to have "decreased OM quality" whereas the glacial periods have "variable but overall higher OM quality". This makes sense when thinking about relative temperature and rates of cycling and, perhaps, the different residence time of OM within the active layer where OM degradation occurs. However, the OM stock sizes between the interglacial and glacial periods must be quite different. Therefore, we might consider the differing consequences of releasing a relatively small amount of "fresh" glacial-era OM compared to relatively large amounts of "degraded" interglacial-era OM. Additionally, regardless of the characteristics of sediment-bound particulate OM, the Woody Layer contains, of course, wood and other plant detritus that will be readily remineralized upon thaw.

We agree that the stock information is a logical and needed next step to continue OM research at the Batagay site. As in this study we focussed on the biomarker approach, testing its applicability to very old permafrost, but we did not yet assess and quantify horizon-specific OM volumes yet. A remote-sensing and UAV-based study is in progress and might provide basic morphological information to allow for OM release estimates from individual stratigraphic (glacial/interglacial) horizons. Furthermore, the interglacial deposits are characterised by a small thickness and consist partly of eroded/reworked material, while the glacial deposits are in situ accumulations and account for more than 80 % of the headwall's vertical extent.

- Related to the above, I think a worthwhile and interesting calculation would be to estimate (even roughly) the relative sizes of C stocks within each of the types of units. If we could estimate the average C stock (i.e., organic C density per unit area) of these

units, could we then also estimate (again, roughly) the amount of C mobilized by the Batagay slump since its formation?

The volume of sediments, C and nutrients mobilised in the Batagay thaw slump is enormous and unique. The quantification of these volumes is important, but, as mentioned above, a separate detailed study is in progress that considers the complex 3D stratigraphy of the Batagay permafrost deposits. With the present study, we aimed to quantify the biogeochemical carbon characteristics, but do not estimate stocks or fluxes as relevant additional data still need to be obtained and analysed. Adding a OM volume assessment taking into account stratigraphy to this biomarker-focused study would substantially inflate the paper size due to the need to explain several additional methods and distract from the core message on OM quality in our view.

- Combined with stock estimates, the authors could incorporate some of the biomarker-based degradation insights to categorize the pools of carbon mobilized as either "pre-processed" or "fresh" to perhaps get some insight into if we expect the mobilized material to be quickly remineralized or simply redeposited downstream. Combining this with knowledge of other thaw slumps could be useful for developing some insights into the consequences of this type of extreme thaw into local (nutrient loading), regional (source of deltaic organic matter), and even global (atmospheric) carbon cycles.

Unfortunately, as we did not perform incubations in the present study, we have no information on the freshness or the "pre-processing" of the OM. Furthermore, as we did not calculate stock estimates as mentioned above, we feel unable to implement this undoubtedly valuable recommendation at this point, but leave it to upcoming research.

- This may be more appropriate for a different article (perhaps one with a stronger focus on cryostratigraphy and geomorphology), but, is the size/scale of Batagay a unique feature? Retrogressive thaw slumps are well-studied and widely documented, but the scale of Batagay is quite impressive. While we can expect that as we warm the Arctic, we will have more thaw-related features, will we expect more Batagay-scale slumps? And, do we think there is anything unique in terms of biogeochemical cycling and/or consequences for local/downstream ecosystems of a single Batagay-scale slump versus multiple, smaller slumps whose total volume of mobilized permafrost might be similar to Batagay?

Rapid permafrost thaw is indeed assumed to accelerate and thus to create more megaslumps in some ice-rich permafrost regions in near future. Multiple studies point at an accelerating rapid degradation of ice-rich permafrost landscapes by thaw slumping, including regions with buried glacial ice but also syngenetic Yedoma ice wedges (Lantz et al., 2008; Lacelle et al., 2010; Kokelj et al., 2017; Lewkowicz and Way, 2019; Runge et al., 2022). In their study of thaw slumps in northwestern Canada, Lacelle et al. (2015) found 189 active slumps of which 10 exceeded 20 ha. Kokelj et al. (2015) referred to slumps as mega slumps when they reached 5-40 ha. Also, recent remote sensing work on thaw slumps (e.g., Kokelj et al., 2015; Runge et al., 2022) suggested that mega slumps (up to 52 ha or larger) are rather rare so far. Therefore, at this

point the Batagay thaw slump is very unique in its size and the largest as far as we know. There might be similarly large or larger slumps that are not researched yet. As the initial disturbance and the onset of the Batagay megaslump are believed to be anthropogenic, it represents however an outstanding example of rapid permafrost thaw that is promoted, but was not originally caused by arctic warming. It would require permafrost modelling for rather detailed local to regional conditions to determine whether large thaw slumps are more likely to form in the future. Here, we hesitate to upscale insights from the Batagay megaslump as its setting and dynamics seem rather unique. Regarding the location of the slump, previous studies reported that thaw slumps inland become more and more important in northwestern Canada in comparison to coastal thaw slumps (Riedlinger and Berkes, 2001; Lewkowicz and Way, 2019).

We added the following sentences in the last chapter of the discussion: "Multiple studies pointed to accelerating rapid degradation of ice-rich permafrost landscapes by thaw slumping, including regions with buried glacial ice but also regions with large syngenetic Yedoma ice wedges (Lantz et al., 2008; Lacelle et al., 2010; Kokelj et al., 2017; Lewkowicz and Way, 2019; Runge et al., 2022). In their study of thaw slumps in northwestern Canada, Lacelle et al. (2015) found 189 active slumps of which 10 exceeded 20 ha. However, recent remote sensing work on thaw slumps (e.g., Kokelj et al., 2015; Runge et al., 2022) suggested that mega slumps (up to 52 ha or larger) are rather rare so far. Therefore, at this point the Batagay thaw slump is very unique in its size and the largest feature as far as we know. As the initial disturbance of the Batagay megaslump is possibly anthropogenic, it represents an outstanding example of rapid permafrost thaw that is promoted, but was not originally caused by arctic warming." (L357)

- The authors note that an unconformity exists between the Lower Sand Unit and the Woody Layer and that the Woody Layer occupies erosional gullies that formed during the last interglacial. While I realize precise dating of accumulation rates is difficult, I would be curious to see an estimate of the amount of carbon mobilized from the Lower Sand Unit due to the warming-induced erosional processes during the last interglacial.

This is a very interesting suggestion and we appreciate this thought. Given the very nature of an unconformity that an entire sediment package is missing due to erosion, the data for such an assessment unfortunately is missing. We actually have no information on how much sediment was removed during past permafrost thaw events. Therefore, we do not see any potential to quantify this with our data from the present study. Given the large age uncertainties of the dating methods applicable to such ancient deposits, i.e. the Lower Sand Unit: >123,200 yr (OSL); 142,800 ± 25,300 yr (OSL) and 210,000 ± 23,000 yr (IRSL; see Murton et al. 2022, https://doi.org/10.1017/qua.2021.27), there is no sensible approach to calculate accumulation rates from the remaining Lower Sand unit, and to capture its material loss by thaw during the Last Interglacial. This may change once there are new and consistent dating results available. Luminescence dating is currently in progress but results will not be available until the end of this year.

---

## Author Comment (AC2)

**Reviewer 2**

Permafrost soils store more than 30% of the global surface organic carbon. The thaw-induced carbon release in the form of greenhouse gases would create a positive feedback to amplify climate warming. In this paper, Jongejans et al., present the valuable records of TOC, TN and multiple lipid biomarkers to assess organic matter quality in a 650-ka ancient permafrost of east Siberia. Such old records cover glacial-interglacial climate variations, which provides a valuable opportunity to explore potential influence of climate changes on permafrost carbon cycling. Their biomarker evidences show higher organic matter decomposition during interglacial periods. This will be very useful for understanding of carbon cycles in permafrost regions as global climate warms. I have no major comments on this valuable paper and hence recommend to accept it after the following minor/moderate comments have been covered.

Thank you for this overall positive assessment of our work! We responded to all comments below. Note: the line numbers refer to the preprint version

L22: Add what lipid biomarkers did you analyze in this study, such as alkanes, fatty acids.
We added "(alkanes, fatty acids and alcohols)" to the abstract as suggested.

L29: Delete "or bacterial". Microbial origin contains bacterial origin.
Changed accordingly.

L36: the world's surface soil carbon?
The global soil estimate (3350 Gt) is based on soils to 3 m (2800 Gt) as well as other pools in deep permafrost (500 Gt) and tropical peatlands (50 Gt; Jackson et al., 2017). Therefore, it is not just the surface soil C, but also includes C of deeply buried deposits. We changed the wording to make this clearer: "The global permafrost region contains roughly half of the world's soil carbon (3350 Gt) and in addition, a large deep permafrost carbon pool (>3 m) which is often not accounted for and its amount is uncertain (~500 Gt) (Strauss et al., 2021)."

L57-58: Biomarker tools as tracing permafrost thaw and carbon cycling are very important in this study. I suggest more previous publications are needed to introduce here. Multiple lipid biomarkers have been applied to sediment records for reconstructing carbon perturbations of permafrost (e.g., Evert et al. 2016, https://doi.org/10.1177/0959683616645942; Yao et al., 2021, https://doi.org/10.1130/G48891.1).
Thank you for your recommendation. We rephrased the sentence and added a few relevant references: "The study of lipid biomarkers has been proven useful in previous work to characterise permafrost OM and carbon cycling as well as tracing permafrost thaw has proven useful in previous work (Zech et al., 2009; Strauss et al., 2015; Elvert et al., 2016; Stapel et al., 2016; Jongejans et al., 2018, 2020; Martens et al., 2020; Bröder et al., 2021; Yao et al., 2021)."

L115: What solvents (including solvent volume) do you use for separation the aliphatic, aromatic and polar NSO?

The MPLC is an instrument for the chromatographic separation of the usually complex extract into fractions of different polarity namely an aliphatic, aromatic and NSO fraction. Due to the instrumental design only n-hexane is used as the main transport medium. During this process, the NSO compounds are separated from the aliphatic and aromatic components by a pre-column. Then the aliphatics are passing through the main column to form the aliphatic fraction, while the aromatics will stick on top of the main column. With the help of a backflush vent the aromatics are eluted from the main column with n-hexane. Subsequently, the NSO compounds are eluted from the pre-column using dichloromethane with 1% methanol. We added now the reference describing the MPLC methods in detail to avoid extending the method description too much.

L127: The first mention of abbreviation "FA" is fatty acids?

Yes. We introduced the abbreviation in line 21 and used the abbreviation throughout the manuscript.

L127: ACL can be also affected by climate changes and resulting alkane degradation, such as temperature and relative humidity. Terrestrial plants tend to produce longer n-alkanes to protect their water loss under higher temperature and drier conditions. Moreover, higher temperature and wetter conditions may facilitate higher microbial activities, may resulting in the faster degradation of organic matter.

Thank you for clarifying this. Indeed, Sachse et al. (2006, https://doi.org/10.1016/j.orggeochem.2005.12.003) found that the ACL index of all analysed *Betula* species increased along a transect from Northern Finland to Southern Italy. It is possible that trees (and with them also other plants) in areas with a longer vegetation period and inhabiting regions with more potential incoming radiation protect their leaves from water loss with longer chain *n*-alkanes. Alternatively, Sachse et al. stated that the observed phenomenon could be caused by evaporative loss of the shorter *n*-alkanes due to increased evaporation southward along their transect. Besides a faster degradation rate, also the annual air temperature could affect *n*-alkane chain length in leaf waxes. Moreover, it has been shown that environmental factors such as moisture and temperature affect the hydrocarbon composition in *Rosmarinus officinalis* leaves on a seasonal scale (Maffei et al., 1993).

We added this limitation in the methods: "Variations of the ACL can be caused by different plant type material and climatic-induced changes of the environmental conditions. For example, different temperature and wetness conditions as well as length of the vegetation periods can influence the long chain n-alkane distribution (e.g., Sachse et al., 2006)" (L129).

L148: Add "may" before "contain".

In line 148, there is no "contain". Do you mean the word "contained"? For that we think that "may" is not the right word, so we added likely instead.

L145-146: Could you show a supplementary figure or table for these correlations?
Following your suggestion, we placed a table with correlation coefficients r in the revised supplements (Table S1). All p-values are below 0.01.

| | alkylcyclopentanes | methylalkanes | diethylalkanes and ethyl-methylalkanes |
|---|---|---|---|
| alkylcyclohexanes | 0.997 | 0.98 | 0.992 |
| alkylcyclopentanes | | 0.974 | 0.986 |
| methylalkanes | 0.974 | | 0.994 |

L227: Please specify what biomolecules or organic indices can give insight into different OM sources.
We added "such as the biomarker indices ACL and $P_{aq}$" to match our work

L240: Change to "higher ACL" and "lower $P_{aq}$".
Changed accordingly

L260: Add a supplementary figure or table for these correlations. And elsewhere.
We placed accordingly a table with correlation coefficients r in the supplements (Table S2). All p-values are below 0.01.

| | ACL | $P_{aq}$ |
|---|---|---|
| CPI | 0.74 | -0.7 |
| ACL | | -0.94 |

L280: Higher ACL does not indicate higher terrestrial source. Higher temperature or drier climate can also lead to higher ACL values.
ACL values around 27 to 31 indicate usually higher plant material. In contrast, lower ACL values might be influenced by more C23/C25 n-alkanes indicating a higher contribution of aquatic OM and/or moss species. Thus, the ACL can provide general information on OM sources. In addition to this, you are right that the ACL is influenced also by climatic induced changes of the environmental conditions. Warmer temperature and more humid conditions seem to impact the ACL to higher ratios. This is sometimes also caused by shifts in the terrestrial plant community (e.g. C3 to C4 plants). However, although there are climate induced variations, the ACL can be used to generally assess the OM type. We explained the parameter in more detail in chapter 3.2 and we used the ACL in combination with the Paq to assess the higher plant and aquatic or moss like character of the OM.

L285: ACL can be affected multiple factors. Please see my comment "L127".
Please see our answer to your comment above

L308: Are there aquatic plants grow around the study area? Higher $P_{aq}$ ratios could also be due to input of mosses - they also produce lots of mid-chain n-alkanes.

That is right, *n*-C23 and *n*-C25 are common both in aquatic macrophytes as well as in (Sphagnum) mosses.. During all stages, ponding water or saturated soil conditions are likely, as the permafrost table prevents percolation below the seasonally thawed uppermost active layer. Ashastina et al. (2017) and Murton et al. (2017) suggested that the modern vegetation also includes mosses. However, there is no Holocene vegetation data from either of those studies (Ashastina et al., 2017; Murton et al., 2017) based on pollen and/or plant macrofossil data. We added in the text that the high Paq could also indicate moss rich OM and included this into the discussion (L130).

L318 and 330: Again, delete "or bacterial". Microbial origin contains bacterial origin.

Changed accordingly.

L346: Change to "rivers".

Changed accordingly.

L357: Add more references (e.g., Evert et al. 2016, https://doi.org/10.1177/0959683616645942; Yao et al., 2021, https://doi.org/10.1130/G48891.1).

We added the suggested manuscript of Yao et al. (2021) as well as a recent manuscript from Bröder et al. (2021) on permafrost C mobilisation from retrogressive thaw slumps.

L359: Please specify what past environments.

We refer to all environments which are not recent. We changed the wording to "palaeo environment".

L370: The impacts of findings should be described as well. E.g. why are these findings important and for whom?

We added the following sentences to the conclusion: "Our biomarker analyses of ancient permafrost sediments contribute to a better understanding of how OM is incorporated and preserved in permafrost deposits during glacial and interglacial periods. Furthermore, it helps to improve our comprehension of possible consequences resulting from future permafrost thaw and OM mobilisation." (L370)

Table 2: Add m/z data of molecular weight, base peak, and characteristic peak of each individual compounds.

We revised Table 2, now including the information on the M+ and characteristic fragments with marked base peak fragments.

---

## Author Comment (AC3)

**Reviewer 3**

This manuscript provides detailed lipid biomarker analyses of exposed permafrost (including ancient deposits) from the recently exposed Batagay Megaslump, East Siberia. I generally found the paper to be a good read and the analyses were appropriate for the sample type, of which the deposit is quite novel and covers an impressive time window (up to 650 ka yr). A number of existing/in review studies have worked on this exposure, so it is good to see one which focuses on organic geochemistry/biomarkers.

To improve the manuscript further the palaeoclimate discussion section would benefit from some revisions by placing each time period more clearly in place with the known climate and environmental conditions at the time (palaeoclimate). Placing the results within a critical framework explaining where these results agree with (or disagree with) existing findings in each time period a little bit more clearly would be helpful and enable the interpretation of any further palaeoclimate interpretations of significance. A table might help here and a little bit of wider reading.

Thank you for your comments and feedback. We appreciate your suggestions on the palaeo-climate discussion. In order to have more focus on the interpretation of the biomarker results in a paleoenvironmental context, we added some more comparisons in the discussion (e.g., L239, L248, L251) to previous studies (Ashastina et al., 2017, 2018; Murton et al., 2017, 2021; Opel et al., 2019; Vasil'chuk et ak., 2020; Courtin et al., in accepted) that focused on the palaeo-aspect. Note: the line numbers refer to the preprint version.

I was also wondering if the group had any compound-specific stable isotope analyses, for example on long chain alkanes or fatty acids to identify any differences between interglacial vegetation and its source/productivity? If they do, then these data should be included.

Unfortunately, we did not perform compound-specific isotope analyses on our samples. Thank you for the suggestion, we will keep this in mind for future work

The discussion would also benefit from a synthesis plot plotting regional data from other palaeobotanical proxies from the site (e.g. Ashastina et al., 2018; Opel et al., 2019) and more widely regionally. This will also assist with improving the palaeoclimate discussion section and show to readers more directly the similarities.

We considered your suggestion, but decided we would rather not incorporate such a table in our manuscript. As mentioned above, we added some more comparisons and linkages to previous palaeo-studies in the discussion, but think that a synthesis table is beyond the scope of this manuscript. A first approach of such a summary of the palaeo-climate findings was made by Opel et al. (2019) (Table 4). A full synthesis of palaeo-finding is furthermore complicated as different authors have worked in different years and parts of the slumps. The stratigraphic and

chronological relation between different sampling years and profiles hasn't been fully established yet, mainly due to rather poor age control beyond the radiocarbon range and because additional luminescence dating is currently in progress but results will not be available until the end of this year.

Finally, I note that the current title is quite descriptive in nature. An alternative approach is to revise the title to make it more impactful and focus on the main finding or outcome from the paper. More interrogation of the palaeoclimate discussion section will help here, but an example could be (from the abstract) to devise a title that focus on terrestrial character of the glacial periods, or the high microbial activity in the interglacial. The potential lability of the Holocene deposit (and its vulnerability) could also be an option as here it is possible to make comparisons with older deposits so effectively.

Thank you for the suggestion. We agree that the title could be less descriptive and instead provide some insights on the conclusions. We changed the title to "Molecular biomarkers in Batagay megaslump permafrost deposits reveal clear  differences in organic matter preservation between glacial and interglacial periods".

**Detailed comments**

31. Do you think that the most recent Holocene deposited permafrost is most vulnerable to warming, leading to OM degradation?

No, from the depth-perspective the active layer deepening is not as important in this location as large amounts of all horizons are mobilised every year through deep permafrost thaw, i.e. the lateral thaw and subsequent thermo-denudation. The dominating factor for permafrost vulnerability clearly is the high ice content (mainly ice wedges) of the glacial age deposits.

36-37. Sentences with the same reference (Strauss et al., 2021) could benefit from merging or rephrasing to improve readability.

Changed accordingly.

74. Perhaps change 'herbs occur' to 'herbs are present'?

Changed accordingly.

96. Change 'the Spring Expedition' to 'a spring expedition'

Changed accordingly.

115-116. What instrument/technique is used for medium pressure liquid chromatography? I think you should include this.

Changed accordingly.

118. "biogeochemical and alkane parameters" Alkane parameters are 'biogeochemical' so rephrase?
We removed "and alkane" as suggested.

Table 1. References to the sources of these indexes could also be included in the table in brackets to enable easy source identification.
We added the references in the table as suggested.

157- 158. Should it read "Here, we found…"?
Changed accordingly.

Figure 2. You display a number of indices. You could also consider calculating and displaying TARFA (Meyers et al., 1993) and CPI for fatty acids (Matsuda and Koyama, 1977).

[Figure]

Yes, we also thought about these parameters, but they didn't add much new aspects to the story. Overall, the CPI of the long *n*-FAs looks rather similar to the CPI of the *n*-alkanes. Glombitza et al. (2009) showed that the CPI of the long FAs decreased with increasing maturation, similar to the CPI of *n*-alkanes. However, this was over the whole range of diagenetic transformation into early catagenesis. Such a broad maturation range cannot be expected for our samples here, which represents immature OM in the initial diagenetic transformation stage. Thus, as for the *n*-alkane CPI, the *n*-FA CPI here also represents rather variable OM sources, which was also observed by Glombitza et al. (2009) for the early diagenetic maturation range. The origin of short chain FAs is not very clear and they can derive from aquatic, terrestrial plant and bacterial biomass. Thus, interpretation is quite difficult and we have other parameters which are more specific such as the IA Index.

The TAR_FA simply shows the ratio between short and long *n*-FAs and could, thus, provide a hint on the ratio between aquatic/bacterial to terrestrial OM. Thus, the very low TAR_FA value in the Woody Layer might suggest that the short FAs are aquatic rather than bacterial, because

the IA Index is relatively high in this layer. However, the TAR_FA is, as explained, not very specific and we decided not to use this parameter in our study.

188. Please double check you are happy with use of the term 'lenses'.
Yes, we prefer to use the same terminology as Murton et al. (2017) ("lenses of woody debris"), Ashastina et al. (2017) ("pronounced lenses") and Opel et al. (2019) ("organic-rich unit … in lenses up to 3 m thick"). We added "organic-rich" in the text.

201. Is the comma (,) necessary after 'Above'?
Yes, because we talk about the Upper Ice Complex and not about the layer above that. For more clarity we removed the word "above".

202 & 214. 'medium high' – how about moderately high?
Changed accordingly

Discussion first paragraph (223-229) – Make clear the differences are seen over contrasting climate periods (glacial-interglacial periods), showing how land-cover varied over different climate regimes.
We added "following different climatic periods (e.g., glacial and interglacial periods)" to the text.

233. 'to some extent' a bit vague.  Can you explain to what extent or specify?
In order to avoid the confusion, we removed "and to some extent also quantitatively".

240. Write 'more terrestrial and less aquatic' to correspond with preceding order of high ACL and low Paq.
Changed accordingly

248. Do we know which microbes were present if abundance and what is their function?  Check with Courtin et al, (in review).
Thank you for your comment. Microbes were present in abundance in every investigated sample from Courtin et al., (accepted) with a core community of Actinobacteria and Haloarchaea in each sample. Still, distinctive patterns were identified between samples. In general, the microbial community consisted mainly of cold adapted and typical permafrost soil Bacteria but for example, in the lower part of the Lower Sand Unit, *Clostridium*, *Nocardioides*, or *Propionibacterium* were well represented suggesting higher degradation and organic matter cycling in the soil. More details can be found in Courtin et al. (accepted). We added this in the discussion (L239, L248, L251).

249. Pollen findings – could these unpublished findings/companion papers data be introduced

here in a summary synthesis plot (most impactful findings) to help with the biomarker comparison?

The pollen data do not cover the same profile as they originate from a reconnaissance sampling campaign in 2017 and therefore cannot be related accordingly.

Discussion of the time periods would be helped by a table showing time period, deposit, references and interpretation to make this section clearer and easier to follow.

As mentioned above we would like to keep the clear focus of the paper. In Figure A1 the results are visualised over depth and unit.

281. Suggest replace 'stronger OM' with greater or 'higher OM'.

Changed accordingly

309. I think the role of diagenesis/preservation in the Holocene deposits could be introduced here.

We are not sure what was meant by this comment.

321. I suggest to change 'stated' to suggested and 'are' to 'could be'.

Changed accordingly.

365 onwards. Given the relatively wet conditions interpreted are you suggesting that the Holocene was a unique interglacial then, geographically at this position, compared with previous interglacial periods?

Thank you for this valuable comment. There were indeed indications from previous work by Kienast et al. (2007; 10.1016/j.gloplacha.2007.07.004) that the last interglacial was characterised by higher temperatures, longer growing seasons and drier conditions. This fits our findings. We added this information to the discussion: "Our findings point to drier conditions during the last interglacial compared to the Holocene, as well as more bioproductivity and microbial degradation, indicating higher temperatures. This fits nicely to the findings of Kienast et al. (2007)." (L335)

[Figure]

**CPI long *n*-fatty acids**

**CPI short *n*-fatty acids**

**TAR *n*-fatty acids**